# SARS-CoV-2 Genomic Characteristics and Clinical Impact of SARS-CoV-2 Viral Diversity in Critically Ill COVID-19 Patients: A Prospective Multicenter Cohort Study

**DOI:** 10.3390/v14071529

**Published:** 2022-07-13

**Authors:** Slim Fourati, Etienne Audureau, Romain Arrestier, Stéphane Marot, Claire Dubois, Guillaume Voiriot, Charles-Edouard Luyt, Tomas Urbina, Julien Mayaux, Anne-Marie Roque-Afonso, Tài Pham, Luce Landraud, Benoit Visseaux, Damien Roux, Raphael Bellaiche, Anne-Sophie L’honneur, Zakaria Ait Hamou, Ségolène Brichler, Stéphane Gaudry, Maud Salmona, Raphaël Clere-Jehl, Elie Azoulay, Laurence Morand-Joubert, Anne-Geneviève Marcelin, Marie-Laure Chaix, Diane Descamps, Armand Mekontso Dessap, Christophe Rodriguez, Jean-Michel Pawlotsky, Nicolas de Prost

**Affiliations:** 1Department of Virology, Hôpitaux Universitaires Henri Mondor, Assistance Publique—Hôpitaux de Paris, 94010 Créteil, France; slim.fourati@aphp.fr (S.F.); christophe.rodriguez@aphp.fr (C.R.); jean-michel.pawlotsky@aphp.fr (J.-M.P.); 2Université Paris-Est-Créteil (UPEC), 94010 Créteil, France; armand.dessap@aphp.fr; 3INSERM U955, Team «Viruses, Hepatology, Cancer», 94010 Créteil, France; 4INSERM U955 Team CEpiA, University Paris-Est-Créteil, 94010 Créteil, France; etienne.audureau@aphp.fr; 5Department of Public Health, Hôpitaux Universitaires Henri Mondor, Assistance Publique—Hôpitaux de Paris, 94010 Créteil, France; 6Médecine Intensive Réanimation, Hôpitaux Universitaires Henri Mondor, Assistance Publique—Hôpitaux de Paris, 94010 Créteil, France; romain.arrestier@aphp.fr; 7Groupe de Recherche Clinique CARMAS, Université Paris-Est-Créteil (UPEC), 94010 Créteil, France; tai.pham@aphp.fr; 8Department of Virology, Assistance Publique–Hôpitaux de Paris, Hôpital Pitié–Salpêtrière, Institut Pierre Louis d’Epidémiologie et de Santé Publique (IPLESP), Sorbonne Université, INSERM U1136, 75013 Paris, France; stephanesylvain.marot@aphp.fr (S.M.); anne-genevieve.marcelin@aphp.fr (A.-G.M.); 9Laboratoire de Virologie, Hôpital Universitaire Saint-Antoine, Institut Pierre Louis d’Epidémiologie et de Santé Publique, Sorbonne Université, INSERM, AP-HP, 75012 Paris, France; claire.dubois@aphp.fr (C.D.); laurence.morand-joubert@aphp.fr (L.M.-J.); 10Médecine Intensive Réanimation, Hôpital Tenon, Assistance Publique—Hôpitaux de Paris, 75020 Paris, France; guillaume.voiriot@aphp.fr; 11Médecine Intensive Réanimation, Assistance Publique—Hôpitaux de Paris, Hôpital Pitié–Salpêtrière, INSERM UMRS_1166-iCAN, Institute of Cardiometabolism and Nutrition, Sorbonne Université, 75013 Paris, France; charles-edouard.luyt@aphp.fr; 12Médecine Intensive Réanimation, Assistance Publique—Hôpitaux de Paris, Hôpital Saint-Antoine, Sorbonne Université, 75571 Paris, France; tomas.urbina@aphp.fr; 13Médecine Intensive Réanimation, Assistance Publique–Hôpitaux de Paris, Hôpital Pitié–Salpêtrière, Sorbonne Université, 75013 Paris, France; julien.mayaux@aphp.fr; 14Laboratoire de Virologie, Hôpital Paul Brousse, Assistance Publique—Hôpitaux de Paris, 94800 Villejuif, France; anne-marie.roque@aphp.fr; 15Service de Médecine Intensive-Réanimation, Assistance Publique—Hôpitaux de Paris, Hôpital de Bicêtre, DMU 4 CORREVE Maladies du Cœur et des Vaisseaux, FHU Sepsis, Groupe de Recherche Clinique CARMAS, 94270 Le Kremlin-Bicêtre, France; 16Equipe d’Epidémiologie Respiratoire Intégrative, CESP, Université Paris-Saclay, UVSQ, Univ. Paris-Sud, INSERM U1018, 94807 Villejuif, France; 17Laboratoire de Microbiologie, Hôpital Louis Mourier, Assistance Publique—Hôpitaux de Paris, 92700 Colombes, France; luce.landraud@aphp.fr; 18Service de Virologie, Hôpital Bichat-Claude Bernard, Assistance Publique—Hôpitaux de Paris, Université de Paris, IAME INSERM UMR 1137, 75018 Paris, France; benoit.visseaux@aphp.fr (B.V.); diane.descamps@aphp.fr (D.D.); 19Service de Médecine Intensive Réanimation, DMU ESPRIT, Hôpital Louis Mourier, Assistance Publique—Hôpitaux de Paris, 92700 Colombes, France; damien.roux@aphp.fr; 20Institut Necker-Enfants Malades (INEM), Department of Immunology, Infectiology and Hematology, INSERM U1151, CNRS UMR 8253, 75015 Paris, France; 21Département d’Anesthésie Réanimations Chirurgicales, Hôpitaux Universitaires Henri Mondor, Assistance Publique—Hôpitaux de Paris, 94010 Créteil, France; raphael.bellaiche@aphp.fr; 22Laboratoire de Virologie, Hôpital Cochin, Assistance Publique—Hôpitaux de Paris, 75014 Paris, France; anne-sophie.lhonneur@aphp.fr; 23Médecine Intensive Réanimation, Hôpital Cochin, Assistance Publique—Hôpitaux de Paris, 75014 Paris, France; zakaria.aithamou@aphp.fr; 24Laboratoire de Virologie, Hôpital Avicenne, Assistance Publique—Hôpitaux de Paris, 93000 Bobigny, France; segolene.brichler@aphp.fr; 25Service de Réanimation, Hôpital Avicenne, Assistance Publique—Hôpitaux de Paris, 93000 Bobigny, France; stephanegaudry@gmail.com; 26Laboratoire de Virologie, Assistance Publique—Hôpitaux de Paris, Hôpital Saint-Louis, Université de Paris, INSERM HIPI, 75010 Paris, France; maud.salmona@aphp.fr (M.S.); marie-laure.chaix@aphp.fr (M.-L.C.); 27Médecine Intensive Réanimation, Hôpital Saint-Louis, Assistance Publique—Hôpitaux de Paris, 75010 Paris, France; raphael.clere-jehl@chru.strasbourg.fr (R.C.-J.); elie.azoulay@sls.aphp.fr (E.A.)

**Keywords:** COVID-19, SARS-CoV-2, variant of concern, acute respiratory failure, intensive care unit

## Abstract

The SARS-CoV-2 variant of concern, α, spread worldwide at the beginning of 2021. It was suggested that this variant was associated with a higher risk of mortality than other variants. We aimed to characterize the genetic diversity of SARS-CoV-2 variants isolated from patients with severe COVID-19 and unravel the relationships between specific viral mutations/mutational patterns and clinical outcomes. This is a prospective multicenter observational cohort study. Patients aged ≥18 years admitted to 11 intensive care units (ICUs) in hospitals in the Greater Paris area for SARS-CoV-2 infection and acute respiratory failure between 1 October 2020 and 30 May 2021 were included. The primary clinical endpoint was day-28 mortality. Full-length SARS-CoV-2 genomes were sequenced by means of next-generation sequencing (Illumina COVIDSeq). In total, 413 patients were included, 183 (44.3%) were infected with pre-existing variants, 197 (47.7%) were infected with variant α, and 33 (8.0%) were infected with other variants. The patients infected with pre-existing variants were significantly older (64.9 ± 11.9 vs. 60.5 ± 11.8 years; *p* = 0.0005) and had more frequent COPD (11.5% vs. 4.1%; *p* = 0.009) and higher SOFA scores (4 [3–8] vs. 3 [2–4]; 0.0002). The day-28 mortality was no different between the patients infected with pre-existing, α, or other variants (31.1% vs. 26.2% vs. 30.3%; *p* = 0.550). There was no association between day-28 mortality and specific variants or the presence of specific mutations. At ICU admission, the patients infected with pre-existing variants had a different clinical presentation from those infected with variant α, but mortality did not differ between these groups. There was no association between specific variants or SARS-CoV-2 genome mutational pattern and day-28 mortality.

## 1. Introduction

As SARS-CoV-2 evolves and new variants continuously emerge worldwide, the sustained monitoring and rapid assessment of genetic changes are required to inform its public health response and health-care management. Since the fall of 2020, the emergence of so-called SARS-CoV-2 “variants of concern” (VOC) and of several “variants of interest” (VOI) appears to have resulted from the adaptation of the virus, combined with strong selective pressures on viral spread in the context of ongoing collective immunization [1]. Many of the substitutions that characterize emerging variants (VOC and VOI) are located in the Spike (S) protein, the protein involved in viral attachment and entry into cells and the target of neutralizing antibodies.

SARS-CoV-2 VOC bearing multiple signature (lineage-defining) deletions and amino-acid substitutions, such as N501Y and/or E484K (variants α (B.1.1.7), β (B.1.351) and γ (P.1)) spread worldwide, including in France, from early 2021 onwards. These changes have been suggested to result in increased infectivity, with 50–74% increased transmissibility for VOC α (B.1.1.7). More recently, variant δ (B.1.617.2) carrying L452R, T478K, and P681R substitutions, with enhanced transmissibility compared to other VOC, has spread and replaced most of the circulating variants in many parts of the world. Beyond transmissibility and immune evasion, the viral factors that explain the severity of infection and the corresponding case-fatality rate in some patients remain unclear. Mutations located outside of the Spike gene in the SARS-CoV-2 genome have been suggested to contribute to enhanced virulence, but these results have not been confirmed in other studies [2,3,4]. Similarly, the specific role of each VOC in the severity of the disease is debated. In a preclinical Syrian golden hamster model, the animals infected with variant α (B.1.1.7) produced significantly higher levels of proinflammatory cytokines than those infected with other variants, but there was no evidence for an altered phenotype [5]. In clinical settings, preliminary analyses of the prognosis in patients infected with variant α (B.1.1.7) have suggested that this variant could be associated with a higher risk of hospital and ICU admission [6,7] and mortality [8,9] compared with infection with other variants. By contrast, other studies found no association between mortality and infection with variant α (B.1.1.7) in patients admitted to hospital [10]. Very few clinical studies have evaluated whether infection with VOC and VOI would modify the prognosis in the subset of patients with severe disease admitted to intensive care units (ICUs) [6].

Most of the prognostic studies describing the role of VOC compared to pre-existing variants are based on PCR-based screening tests, such as S-gene molecular diagnostic assay failure (SGTF) for variant α (B.1.1.7) [6,8,11], which did not discriminate between different variants sharing the same pattern of mutations and could not evaluate the association of mutational hotspots with clinical outcomes. Genomic surveillance by means of full-length viral-genome sequencing has become critical to identifying circulating variants and the evolution of SARS-CoV-2 genomes over time. Different variants, including VOC and VOI, circulate at the same time, and their effects on disease severity and on the prognosis of severe forms remain unknown. At a mutational level, the presence of a single substitution (e.g., E484K) or a panel of mutations/deletions could specifically modulate immune response, with a potential impact on the clinical course of the disease [12].

In this study, we aimed to characterize the genetic diversity of the SARS-CoV-2 variants involved in severe COVID-19 cases, in a population of patients hospitalized in ICUs for acute respiratory failure following severe SARS-CoV-2 infection between October 2020 and May 2021, i.e., before the emergence and spread of variant δ (B.617.2) in France. We further aimed to unravel the relationships between specific viral mutations/mutational patterns and the clinical outcomes of COVID-19 in these patients.

## 2. Materials and Methods

### 2.1. Study Design and Patients

This is a prospective multicenter observational cohort study. Patients admitted between 1 October 2020 (week 40/20) and 30 May 2021 (week 21/21) to one of the 11 participating ICUs from the hospitals in the Greater Paris area and included in the ANTICOV study (NCT04733105) were eligible for study inclusion. Inclusion criteria were as follows: age ≥18 years, SARS-CoV-2 infection confirmed by a positive reverse-transcriptase–polymerase-chain-reaction (RT–PCR), patient admitted to the ICU for acute respiratory failure (SpO2 ≤ 90% and need for supplemental oxygen or any kind of ventilator support), and patient or next of kin informed of study inclusion. Patients with SARS-CoV-2 infection but no acute respiratory failure or with a PCR cycle threshold (Ct) > 32 in nasopharyngeal samples were not included in the study. The study was approved by the Comité de Protection des Personnes Nord-Ouest IV (N° EudraCT/ID-RCB: 2020-A03009-30). Informed consent was obtained from all patients or their relatives.

Demographics and clinical and laboratory variables were recorded upon ICU admission and during ICU stay. Patients’ frailty was assessed using the Clinical Frailty Scale [13]. The severity of the disease upon ICU admission was assessed using the World Health Organization (WHO)’s 10-point ordinal scale [14], the sequential organ failure assessment (SOFA [15]) score, and the simplified acute physiology (SAPS) II score [16]. Acute respiratory distress syndrome (ARDS) was defined according to the Berlin definition [17]. The primary clinical endpoint of the study was day-28 mortality.

#### SARS-CoV-2 Genome Sequence Analysis

The full-length SARS-CoV-2 genomes from all included patients were sequenced by means of next-generation sequencing. Briefly, viral RNA was extracted from nasopharyngeal swabs in viral transport medium using NucliSENS^®^ easyMAG kit on EMAG device (bioMérieux, Marcy-l’Étoile, France). Sequencing was performed with the Illumina COVIDSeq Test (Illumina, San Diego, CA, USA), which uses 98-target multiplex amplifications along the full SARS-CoV-2 genome. The libraries were sequenced with NextSeq 500/550 High-Output Kit v2.5 (75 Cycles) on a NextSeq 500 device (Illumina). The sequences were demultiplexed and assembled as full-length genomes by means of the DRAGEN COVIDSeq Test Pipeline on a local DRAGEN server (Illumina). Lineages and clades were interpreted using Pangolin and NextClade, before being submitted to the GISAID database (https://www.gisaid.org; accessed on 1 June 2021; the GISAID IDs of the sequenced sampled are listed in the Appendix A). Phylogeny was performed after full-length genome alignment with Muscle v3.8.31 (maximum-likelihood model GTR + I; 1000 bootstrap replicates), by means of IQ-Tree v1.3.11.1 and iTOL.

### 2.2. Statistical Analysis

Descriptive results are presented as means (±standard deviation [SD]) or medians (1st–3rd tertials) for continuous variables, and as numbers with percentages for categorical variables. Two-tailed *p*-values < 0.05 were considered statistically significant. Unadjusted comparisons between the main variants were performed using ANOVA or Kruskal–Wallis tests for global comparisons of continuous variables, and Chi-squared or Fisher’s exact tests for categorical variables, as appropriate. In case of global significance, post hoc pairwise comparisons were performed using t-tests or Mann–Whitney tests for continuous variables, and Chi-squared or Fisher’s exact tests for categorical variables, as appropriate, applying a Sidak correction to account for multiple testing. Adjusted analyses of the association between variants or mutations and 28- and 90-day mortality relied on logistic regression models, systematically adjusting for age, SOFA score at admission, gender and dexamethasone treatment, computing adjusted odds ratios (ORs) along with their 95% confidence intervals. Analyses were performed using Stata V16.0 statistical software (StataCorp, College Station, TX, USA), and R 3.6.3 (R Foundation for Statistical Computing, Vienna, Austria).

## 3. Results

Between 1 October 2020 and 30 May 2021, 845 patients were admitted to 11 participating ICUs. Of these patients, 737 had at least one positive SARS-CoV-2 RNA RT–PCR, according to a nasopharyngeal swab sample in the hospital; 413 patients with a Ct ≤ 32 and a nasopharyngeal sample maintained at −80 °C that could be used for full-length viral genome sequence analysis were included in this study. Among these 413 patients, 183 (44.3%) were infected with so-called “pre-existing” variants, i.e., variants circulating before the emergence of variant α, 197 (47.7%) were infected with variant α (B.1.1.7), and 33 (8.0%) were infected with other variants, including β (B.1.351) (n = 19, 4.6%), γ (P.1) (n = 2, 0.5%), and other variants of interest (n = 12, 2.9%). Figure 1 illustrates the time course of the emerging SARS-CoV-2 variants during the study period (see Appendix A for the detailed number of collected samples per week). Variant α (B.1.1.7) was first detected during the last week of 2020 in 11.1% of the patients, became predominant during the fourth week of January 2021 (51.9% of the patients) and remained so until the last week of May (100% of patients), which corresponded to the end of the inclusion period.

### 3.1. SARS-CoV-2 Variant Association with Clinical Phenotypes at ICU Admission

The patients infected with pre-existing variants were older and had significantly more frequent chronic kidney disease and COPD than those infected with other variants (Table 1). Other comorbidities, including diabetes, peripheral vascular disease, solid cancer, and hypertension followed the same trend, although the differences were not statistically different.

There were also marked differences between the variant groups regarding the severity of disease at ICU admission. Indeed, the patients infected with preexisting variants exhibited significantly higher severity-of-illness scores, with not only more severe respiratory disease, as assessed by the respiratory component of the SOFA score, but also more frequent extra-pulmonary organ failure compared with the patients infected with variant α (B.1.1.7) and other emerging variants (Table 1 and Figure 2). As a result, the patients infected with the pre-existing variants received less frequent high-flow oxygen therapy and more frequent extracorporeal membrane oxygenation (ECMO) support than the other patients within the first 24 h of ICU admission, while they met the ARDS-definition criteria more frequently.

### 3.2. Relationship between SARS-CoV-2 Variants and Patient Outcomes

Although the patients infected with pre-existing variants were older, had more comorbidities, and had more severe disease at ICU admission, they did not show different outcomes from those infected with the other variants. Overall, the need for invasive mechanical ventilation or ECMO support did not significantly differ between the different variant groups, nor did the duration of these supports. There were also no significant differences between groups regarding extra-pulmonary organ support (i.e., vasopressors and renal replacement therapy) during ICU stay (Table 2). The patients with pre-existing variants received less dexamethasone and tocilizumab than the others, likely reflecting the changes in practice during the study period. As shown in Table 2, the day-28 and day-90 mortality rates were not statistically different between the groups.

### 3.3. Relationship between SARS-CoV-2 Spike Mutations and Mortality

Pre-existing and emerging variants (VOC or VOI) are characterized by multiple lineage-specific deletions and amino-acid substitutions. Spike mutations undergo evolutive convergence at several signature sites. The spike mutations at sites identified as undergoing convergent mutational evolution were selected a priori. There was no significant relationship between any of these mutations (n = 17) and day-28 mortality in univariate analysis (Figure 3). The selected mutations were also included in multivariate logistic regression models to determine their relationship with day-28 (Table 3) and day-90 (Appendix A) mortality, adjusting for age, gender, SOFA score at ICU admission, and dexamethasone treatment. None of them were significantly associated with mortality at either time point. The variant status was not associated with mortality either.

### 3.4. Relationship between SARS-CoV-2 Gene-Mutation Hotspots and Mortality

Overall, 1017 non-synonymous mutations (including 953 amino acid substitutions, 52 deletions, and 11 insertions) were detected in full-length viral genomes in at least one variant. There were no significant relationships between any of these mutations and day-28 mortality in the univariate analysis (Table 3). Although 11 mutations were found to be associated with mortality (*p* < 0.05), the number of patients harboring these mutations was considered too small (range: 2–6) to derive conclusions.

We then focused on the mutations previously reported to correlate with worse clinical outcomes [2,4,18]. We found that Q57H in Orf3a (but not other mutations, including P25L in Orf3a and S194L, R203K, or G204R in N) was highly prevalent in our cohort (31.8%). Q57H was more prevalent in patients who were dead at day 28 (35.6%) than in those who were still alive, but the difference did not reach significance (29.0%; *p* = 0.196). A phylogenetic tree (Supplementary Appendix A) shows that the Q57H mutation is mainly harbored by variants from the lineages B.1.160 and B.1.351.

## 4. Discussion

The full-length SARS-CoV-2 genomes of 413 critically ill COVID-19 patients from 11 ICUs, who were predominantly infected with pre-existing and α (B.1.1.7) variants, were sequenced, and the relationship between the viral sequences and their clinical presentation and outcomes was studied. The main results of this study are as follows. (i) Compared with the other patients, the patients infected with the α (B.1.1.7) variant showed a different clinical phenotype at ICU admission, characterized by younger age, fewer comorbidities, and less-severe clinical presentation. (ii) Despite these different initial clinical presentations, there were no significant differences in day-28 and day-90 mortality between the different SARS-CoV-2 variant groups. (iii) There was no statistically significant relationship between the presence of any of the 17 relevant spike substitutions and deletions selected a priori and mortality. (iv) A comprehensive full-length SARS-CoV-2 genome sequence analysis exploring all the mutations found failed to identify a relationship between the presence of any of them and day-28 mortality.

The patients infected with variants that emerged prior to variant α at the end of 2020 showed significant clinical differences, including older age and more comorbidities, when compared with the patients infected with other variants, particularly variant α (B.1.1.7), a result that was in keeping with those from previous cohorts [6,10,11]. Importantly, our results show differences in the clinical severity of the disease, resulting in different organ support requirements during the first 24 h of management in the ICU, in the patients infected with different SARS-CoV-2 variants. Indeed, the patients infected with pre-existing variants exhibited more severe pulmonary and non-pulmonary organ failure than those infected with variant α (B.1.1.7). Surprisingly, although the severity-of-illness scores on admission (i.e., SAPS II [16] and SOFA [15]) predicted higher mortality in the patients infected with the pre-existing variants, there was no significant difference in their mortality rates compared to the patients infected with variant α (B.1.1.7). This finding, together with the fact that the patients infected with variant α (B.1.1.7) had similar organ support requirements during their ICU stay (Table 2), indicates that the severity of the disease and the time course of organ failures were delayed in the patients infected with variant α compared to the patients infected with the pre-existing variants. Whether this different time course should be ascribed to host-related factors (e.g., age, comorbidities), a different pathogenesis/virulence of the variant, or differences in the type of patient management, is unclear. Previous large-scale data have suggested that patients infected with variant α (B.1.1.7), identified using the SGTF proxy had a higher risk of dying [6,8,11]. However, in a study by Frampton et al., which included 341 hospitalized patients, no association was found between the severity of the disease, death, and the viral lineage [10]. Interestingly, following SARS-CoV-2-infected patients with different disease severities through the pathway of disease management, Grint et al. reported a weakening association between variant-α (B.1.1.7) infection and mortality, which was highest in the primary-care population, lower in the hospitalized population, and non-significant in patients admitted in the ICU [11]. Indeed, other determinants of COVID-19 mortality, including male gender, age, and associated comorbidities, have been shown to have a major impact on outcomes [19]. Our results thus do not contradict those of previous studies showing an increased association of mortality with variant α (B.1.1.7) in the community [6,8]. The results are in keeping with data specific to ICU patient populations [6,11].

Spike mutations have dominated SARS-CoV-2 variant research due to concerns regarding the enhancement of viral replication and transmission via ACE-2 receptor binding and/or lower sensitivity to the action of naturally or vaccine-induced neutralizing antibodies. By contrast, little attention has been paid to variant-specific mutations in other viral proteins. Nevertheless, such mutations could be associated with different clinical outcomes. Previous large-scale studies based on the GISAID database and in vitro experiments suggested that amino-acid substitutions in the N and Orf3a regions could be associated with more severe disease. To study the potential link between SARS-CoV-2 genetic diversity outside of the spike protein and severe COVID-19-case outcomes, we analyzed all non-synonymous mutations in the full-length SARS-CoV-2 genomes and not only assessed the relationship between variant status and mortality, as already performed by others [6,8,10,11], but also performed a mutational analysis of all the viral genes. None of the relevant pre-selected spike substitutions/deletions were found to be associated with mortality, either in univariate or in multivariate analyses. When extending the analysis to all the viral genes, we found 11 mutations that were harbored by a very small number of patients (n ≤ 6) with a statistically significant difference, but without clinically relevant meaning. Together, our results indicate that in the patients with the most severe forms of COVID-19 who required ICU admission, there was no mutational pattern associated with mortality.

### Limitations and Strengths of the Study

We acknowledge that our study has several limitations. The number of patients included was limited and, thus, the statistical power may have been too weak to show some between-group differences. However, there was no clear trend regarding associations between variant groups/mutations and mortality, suggesting that increasing the number of patients in the cohort would not have changed the results. Changes in management practices occurred during the study period, with more patients infected with variant α (B.1.1.7) and other variant groups receiving dexamethasone, with a possible impact on mortality [20]. This potential bias was accounted for in the multivariate analysis. We did not use a population-based database, precluding the study of relationships between the effects of the COVID-19 epidemic and risk of ICU death according to variant type. Additionally, the period of inclusion of the present study began on October 1, 2020, which corresponded in France to the beginning of the second COVID-19 wave, a period when ICU admission policies, ICU patient load and demand, and COVID-19 management strategies were more homogeneous among centers than during the first COVID-19 wave. We also did not record SARS-CoV-2 vaccination status as there was no anti-SARS-CoV-2 vaccine available at the time the study started. However, on May 1 2021, when the inclusion period ended, less than 10% of the French population had been fully vaccinated (https://covidtracker.fr/vaccintracker/; accessed on 1 June 2022), implying that the proportion of vaccinated patients in this cohort of critically ill patients was very low.

Our study also had strengths, including the constitution of a prospective multicenter cohort of well-phenotyped critically ill patients, and the fact that we performed full-length SARS-CoV-2 genome sequencing by means of up-to-date technology.

## 5. Conclusions

The patients with the pre-existing variants had a different clinical presentation from the patients infected with variant α (B.1.1.7) and other variants at ICU admission, characterized by older age, more comorbidities, and a more severe clinical presentation. However, there were no day-28 or day-90 mortality differences between the different groups. We found no association between the variant status or any mutational pattern in SARS-CoV-2 viral genes and mortality.

## Figures and Tables

**Figure 1 viruses-14-01529-f001:**
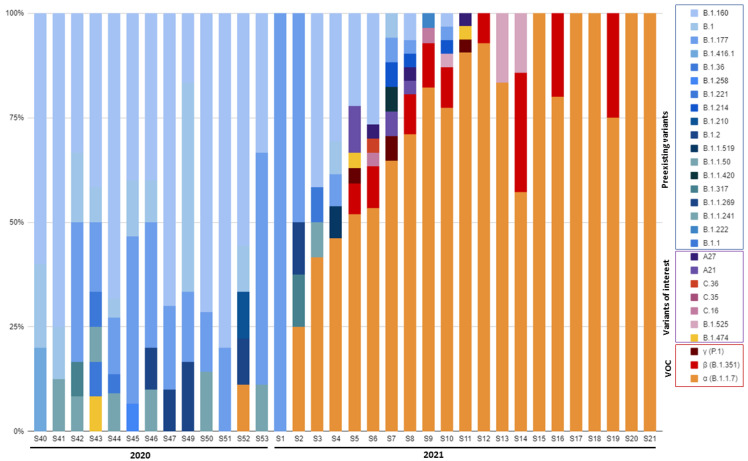
Time course of emerging SARS-CoV-2 variants during the study period. VOC, variants of concern.

**Figure 2 viruses-14-01529-f002:**
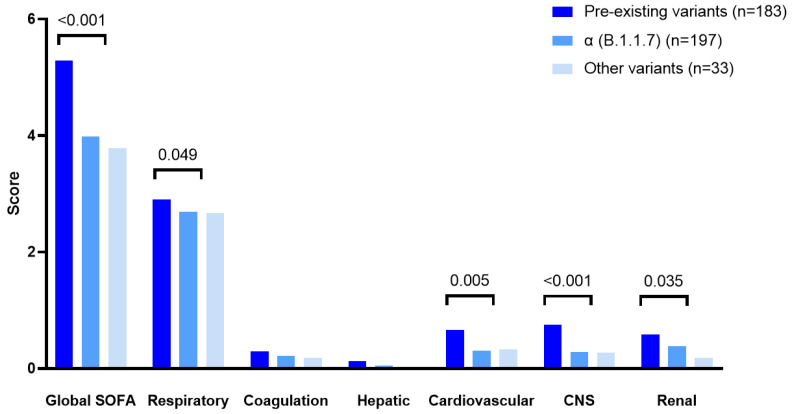
SOFA score and its organ system components in patients with pre-existing, α (B.1.1.7), or other variants. The *p* values are from ANOVA.

**Figure 3 viruses-14-01529-f003:**
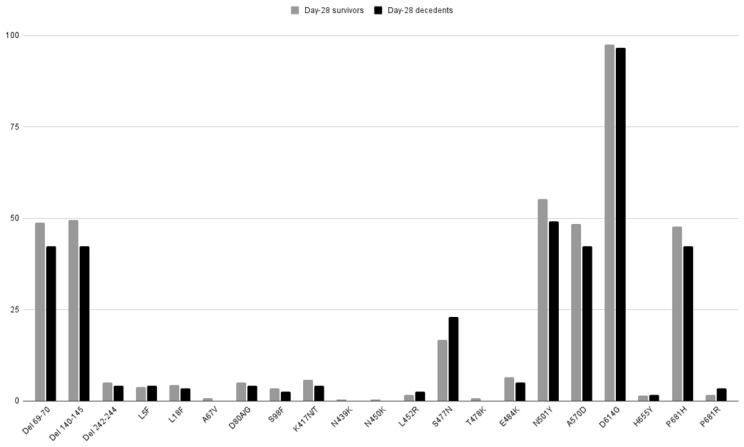
Frequency of preselected SARS-CoV-2 mutations in spike according to day-28 mortality status. There was no statistical difference in any of the comparisons performed (gray bars indicate survivors, black bars indicate deceased).

**Table 1 viruses-14-01529-t001:** Characteristics of patients with severe SARS-CoV-2 infection (n = 413) at the time of intensive-care-unit admission, according to SARS-CoV-2 variants.

	Pre-Existing Variants	Alpha (B.1.1.7)	Other Variants	*p*-Values			
	N = 183	N = 197	N = 33	Global Comparison	*Pre-Existing* vs. *Alpha* ^a^	*Pre-Existing* vs. *Others* ^a^	*Alpha* vs. *Others* ^a^
** *Demographics and comorbidities* **							
Gender, male	124 (67.8%)	140 (71.1%)	18 (54.5%)	0.165	-	-	-
Age, years	65.0 (±11.9)	60.5 (±11.8)	62.2 (±10.7)	**0.001**	**0.001**	0.531	0.823
Diabetes	67 (36.6%)	59 (29.9%)	7 (21.2%)	0.141	-	-	-
Hypertension	108 (59.0%)	98 (49.7%)	16 (48.5%)	0.159	-	-	-
Peripheral vascular disease	28 (15.3%)	19 (9.6%)	1 (3.0%)	0.075	-	-	-
Chronic heart failure	19 (10.4%)	20 (10.2%)	2 (6.1%)	0.851	-	-	-
Chronic kidney disease	34 (18.6%)	23 (11.7%)	1 (3.0%)	**0.025**	0.178	0.063	0.517
Cirrhosis	3 (1.6%)	4 (2.0%)	0 (0.0%)	1.000	-	-	-
Cancer	15 (8.2%)	8 (4.1%)	1 (3.0%)	0.180	-	-	-
HIV infection	1 (0.5%)	3 (1.5%)	1 (3.0%)	0.274	-	-	-
Corticosteroids	14 (7.7%)	9 (4.6%)	1 (3.0%)	0.445	-	-	-
COPD	21 (11.5%)	8 (4.1%)	2 (6.1%)	**0.021**	**0.021**	0.904	0.953
Tobacco	26 (14.2%)	23 (11.7%)	4 (12.1%)	0.769	-	-	-
BMI, kg/m^2^	30.2 (±6.8)	29.8 (±6.6)	30.2 (±6.0)	0.857	-	-	-
Clinical frailty scale	3.0 (2.0;4.0)	3.0 (2.0;4.0)	3.0 (2.0;3.0)	0.069	-	-	-
** *Disease severity upon ICU admission and biological features* **							
WHO 10-point scale	6 (6;8)	6 (6;8)	6 (6;8)	0.114	-	-	-
SAPS II score	37 (30;50)	30 (24;40)	32 (24;41)	**0.000**	**0.000**	**0.025**	0.971
SOFA score	4 (3;8)	3 (2;4)	4 (2;4)	**0.000**	**0.000**	0.156	0.935
Blood neutrophils, G/L	7.9 (5.3;11.3)	6.3 (4.6;9.1)	8.7 (5.8;12.2)	**0.006**	**0.017**	0.737	0.061
Blood lymphocytes, G/L	0.6 (0.4;0.9)	0.7 (0.4;0.8)	0.8 (0.6;1.0)	0.217	-	-	-
Blood urea level, mM	8.4 (6.0;12.5)	6.5 (4.9;10.2)	6.3 (5.5;8.7)	**0.000**	**0.000**	**0.038**	0.999
D-dimers, ng/mL	1441 (771;2614)	1310 (931;2183)	1300 (799;2030)	0.917	-	-	-
Bacterial coinfection	25 (13.7%)	22 (11.2%)	2 (6.1%)	0.494	-	-	-
** *Organ support and management during the first 24 h ^b^* **							
Oxygen	5 (2.7%)	6 (3.0%)	4 (12.1%)	**0.045**	1.000	0.095	0.114
High-flow oxygen	105 (57.4%)	140 (71.1%)	23 (69.7%)	**0.017**	**0.016**	0.458	0.998
NIV/C-PAP	55 (30.1%)	51 (25.9%)	12 (36.4%)	0.392	-	-	-
Invasive MV	88 (48.1%)	78 (39.6%)	12 (36.4%)	0.178	-	-	-
Prone position	79 (43.2%)	92 (46.7%)	15 (45.5%)	0.786	-	-	-
ECMO	24 (13.1%)	9 (4.6%)	3 (9.1%)	**0.010**	**0.010**	0.989	0.770
ARDS criteria	147 (80.3%)	136 (69.0%)	21 (63.6%)	**0.018**	**0.035**	0.098	0.901
Vasopressors	35 (19.1%)	34 (17.3%)	4 (12.1%)	0.654	-	-	-
Antibiotics	123 (67.2%)	115 (58.4%)	19 (57.6%)	0.175	-	-	-

^a^ Pairwise comparisons applying a Sidak correction to account for multiple testing; ^b^ more than one type of respiratory support may have been used per patient during the first 24 h; thus the total may be more than 100%. COPD: chronic obstructive pulmonary disease; BMI: body mass index; SAPS: simplified acute physiology score; SOFA: sequential organ failure assessment; NIV: non-invasive ventilation; C-PAP: continuous positive airway pressure; MV: mechanical ventilation; ECMO: extracorporeal membrane oxygenation; ARDS: acute respiratory distress syndrome.

**Table 2 viruses-14-01529-t002:** Intensive-care management and outcomes of patients with severe SARS-CoV-2 infection (n = 413) during intensive-care-unit stay, according to SARS-CoV-2 variants.

	Preexisting Variants	Alpha (B.1.1.7)	Other Variants	*p*-Values			
	N = 183	N = 197	N = 33	Global Comparison	*Preexisting* vs. *Alpha* ^a^	*Preexisting* vs. *Others* ^a^	*Alpha* vs. *Others* ^a^
Invasive MV	125 (68.3%)	120 (60.9%)	22 (66.7%)	0.312	-	-	-
MV duration, days	16 (9;27)	14 (8;23)	16.5 (10;30)	0.362	-	-	-
ECMO support	29 (15.8%)	19 (9.6%)	6 (18.2%)	0.133	-	-	-
Duration of ECMO, days	17.0 (6.0;31.0)	12.0 (4.0;17.0)	34.5 (8.0;55.0)	0.186	-	-	-
Vasopressor support	87 (47.5%)	94 (47.7%)	17 (51.5%)	0.912	-	-	-
Duration of vasopressors, days	7 (3;15)	9 (4;16)	9 (3;14)	0.862	-	-	-
RRT	47 (25.7%)	53 (26.9%)	7 (21.2%)	0.784	-	-	-
Pulmonary thrombosis	10 (5.5%)	11 (5.6%)	1 (3.0%)	1.000	-	-	-
VAP	88 (48.1%)	85 (43.4%)	15 (45.5%)	0.654	-	-	-
Dexamethasone	143 (78.1%)	168 (85.3%)	31 (93.9%)	**0.038**	0.199	0.101	0.444
Tocilizumab	6 (3.3%)	25 (12.7%)	4 (12.1%)	**0.002**	**0.003**	0.139	1.000
Day-28 mortality	57 (31.1%)	51 (26.2%)	10 (30.3%)	0.550	-	-	-
Day-90 mortality	74 (40.4%)	61 (31.6%)	13 (39.4%)	0.189	-	-	-

^a^ Pairwise comparisons applying a Sidak correction to account for multiple testing. MV: mechanical ventilation; ECMO: extracorporeal membrane oxygenation; RRT: renal replacement therapy; VAP: ventilator-acquired pneumonia.

**Table 3 viruses-14-01529-t003:** Association between SARS-CoV-2 variants, relevant mutations ((substitutions and deletions (Del)) in spike selected a priori and day-28 mortality by multivariable logistic regression analysis.

		All Patients	Day-28 Survivors	Day-28 Non-Survivors	aOR ^a^ (95% CI)	*p*-Value
		N = 411	N = 293	N = 118		
**Variants**	Alpha	195 (47.4%)	144 (49.1%)	51 (43.2%)	1 (ref)	0.347
	Pre-existing variant	183 (44.5%)	126 (43.0%)	57 (48.3%)	0.75 (0.45;1.25)	0.263
	Others	33 (8.0%)	23 (7.8%)	10 (8.5%)	1.31 (0.55;3.13)	0.549
N501Y		220 (53.5%)	162 (55.3%)	58 (49.2%)	1.29 (0.80;2.11)	0.299
N501Y x Alpha ^b^	None	191 (46.5%)	131 (44.7%)	60 (50.8%)	1 (ref)	0.565
	N501Y	25 (6.1%)	18 (6.1%)	7 (5.9%)	1.45 (0.53;3.97)	0.473
	N501Y and Alpha	195 (47.4%)	144 (49.1%)	51 (43.2%)	1.28 (0.77;2.11)	0.342
Del 69–70		193 (47.0%)	143 (48.8%)	50 (42.4%)	1.22 (0.75;1.98)	0.426
Del 140–145		195 (47.4%)	145 (49.5%)	50 (42.4%)	1.20 (0.74;1.95)	0.468
Del 242–244		20 (4.9%)	15 (5.1%)	5 (4.2%)	1.34 (0.44;4.09)	0.607
L5F		16 (3.9%)	11 (3.8%)	5 (4.2%)	1.02 (0.32;3.27)	0.978
L18F		17 (4.1%)	13 (4.4%)	4 (3.4%)	1.19 (0.34;4.10)	0.788
D80A/G		20 (4.9%)	15 (5.1%)	5 (4.2%)	1.21 (0.40;3.66)	0.733
S98F		13 (3.2%)	10 (3.4%)	3 (2.5%)	0.76 (0.18;3.17)	0.703
K417N/T		22 (5.4%)	17 (5.8%)	5 (4.2%)	1.02 (0.34;3.02)	0.976
L452R		8 (1.9%)	5 (1.7%)	3 (2.5%)	1.57 (0.33;7.41)	0.569
S477N		76 (18.5%)	49 (16.7%)	27 (22.9%)	0.79 (0.43;1.45)	0.447
E484K		25 (6.1%)	19 (6.5%)	6 (5.1%)	1.06 (0.38;2.94)	0.908
A570D		192 (46.7%)	142 (48.5%)	50 (42.4%)	1.23 (0.76;2.00)	0.406
D614G		400 (97.3%)	286 (97.6%)	114 (96.6%)	0.65 (0.16;2.68)	0.550
H655Y		6 (1.5%)	4 (1.4%)	2 (1.7%)	1.07 (0.17;6.59)	0.946
P681H		190 (46.2%)	140 (47.8%)	50 (42.4%)	1.29 (0.79;2.09)	0.310
P681R		9 (2.2%)	5 (1.7%)	4 (3.4%)	2.72 (0.61;12.05)	0.189

^a^ Multivariable analysis adjusted for age, SOFA score at admission, gender, and dexamethasone treatment; ^b^ This variable is from the interaction of the variable «alpha variant » and the variable « N501Y mutation»; aOR: adjusted odds ratio; 95% CI: 95% confidence interval.

## Data Availability

Original data are available on request from the corresponding author.

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
