# Peer review of "SARS-CoV-2 Genomic Characteristics and Clinical Impact of SARS-CoV-2 Viral Diversity in Critically Ill COVID-19 Patients: A Prospective Multicenter Cohort Study"

_viruses, 2022, doi:10.3390/v14071529_

Round 1

Reviewer 1 Report

The authors aimed to characterize the genetic diversity of SARS-CoV-2 variants isolated from patients with severe COVID-19 and unravel the relationships between specific viral mutations/mutational patterns and clinical outcomes. They have found that : (i) as compared with others, patients infected with pre-existing variants showed a different clinical phenotype at ICU admission, characterized by older age, more comorbidities, and a more severe clinical presentation; (ii) despite these different initial clinical presentations, there were no significant differences in day-28 and day-90 mortality between the different SARS-CoV-2 variant groups; (iii) there was no statistically significant relationship between the presence of any of 17 relevant spike substitutions and deletions selected a priori and mortality; and (iv) a comprehensive full-length SARS-CoV-2 genome sequence analysis exploring all found mutations failed to identify a relationship between the presence of any of them and day-28 mortality.

Several minor suggestions:

1.     In tabel2, [Table 2] between lines 242 and 243 is suggested to remove.

2.     In table 3, [Variant of origin] is suggested to be [pre-existing variant].

3.     Line 229, [Relationship between SARS-CoV-2 Spike substitutions and deletions and mortality] may change to [Relationship between SARS-CoV-2 Spike mutations and mortality].

4.     Line 252, [Relationship between SARS-CoV-2 spike and other gene mutation hotspots and mortality] may change to [Relationship between SARS-CoV-2 gene mutation hotspots and mortality].

Author Response

The authors aimed to characterize the genetic diversity of SARS-CoV-2 variants isolated from patients with severe COVID-19 and unravel the relationships between specific viral mutations/mutational patterns and clinical outcomes. They have found that : (i) as compared with others, patients infected with pre-existing variants showed a different clinical phenotype at ICU admission, characterized by older age, more comorbidities, and a more severe clinical presentation; (ii) despite these different initial clinical presentations, there were no significant differences in day-28 and day-90 mortality between the different SARS-CoV-2 variant groups; (iii) there was no statistically significant relationship between the presence of any of 17 relevant spike substitutions and deletions selected a priori and mortality; and (iv) a comprehensive full-length SARS-CoV-2 genome sequence analysis exploring all found mutations failed to identify a relationship between the presence of any of them and day-28 mortality.

Authors : We thank the reviewer for appraising our work and for raising constructive comments.

Several minor suggestions:

  1. In tabel2, [Table 2] between lines 242 and 243 is suggested to remove.

Authors : Modified as requested.

  1. In table 3, [Variant of origin] is suggested to be [pre-existing variant].

Authors : Modified as requested.

  1. Line 229, [Relationship between SARS-CoV-2 Spike substitutions and deletions and mortality] may change to [Relationship between SARS-CoV-2 Spike mutations and mortality].

Authors : Modified as requested.

  1. Line 252, [Relationship between SARS-CoV-2 spike and other gene mutation hotspots and mortality] may change to [Relationship between SARS-CoV-2 gene mutation hotspots and mortality].

Authors : Modified as requested.

Reviewer 2 Report

This work describes a multicenter clinical study with the aim to investigate correlations between clinical severity of COVID-19 and mutational patterns of SARS-CoV-2 genomes. The amount of clinical and genomic data from patients admitted at ICU in different hospitals is remarkable. However, there are some fundamental information that the authors did not report and that are essential for their scope.

The first essential information that the author did not mention in their analysis was the vaccination status of the patients involved in the study. During the data collection of the present study (between October 2020 and 30 May 2021), the number of people vaccinated with at list one dose in France increased from 0% to 38,7%. The efficacy of vaccines in preventing mortality is well documented in many works, so the information of vaccination status of patients involved in the study must be considered in the multivariate analysis.

The second essential information that was not reported in this work is the total number of SARS-CoV-2 positive patients. Previous works (like Patone et al., The Lancet Infectious Diseases 2021) observed that patients with variant “alpha” were at increased risk of ICU admission and 28-day mortality; however considering only patients that received critical care, mortality appeared to be independent of virus strain. To better investigate the relationship between variants and clinical outcomes it should be necessary to also evaluate the ratio between the number of deaths and the number of infected people, like Davies et al. Nature 2022, or Funk et al. EUROSURVEILLANCE 2021.

Specific comments:

Line 85: which are “the substitutions selected so far”? By whom? Please explain better

Line 107: Other variants have been also investigated, please cite at list  “Funk et al. EUROSURVEILLANCE 2021”

Lines 159-161 : authors reported the methods of a phylogenetic analysis that is not reported in the results. Please report the SARS-CoV-2 phylogenetic tree: the mutations found with a statistically significant difference are harbored by SARS-CoV-2 genomes that are phyogenetically linked?

Line 186 : please add the list of GISAID IDs of the sequenced samples in supplementary.

Figure 1: please also report the number of samples collected every week. If it not possible to report these data into the same figure, please add another figure as supplementary.

Lines 268-270 : this observation could be inverted: ICU patients infected with variant alpha and were characterized by a younger age and less comorbidities; it is possible to conclude that alpha variant is associated with a severe disease in younger people? Can be better investigated the correlation between mutational patterns and age of ICU admission?

Author Response

This work describes a multicenter clinical study with the aim to investigate correlations between clinical severity of COVID-19 and mutational patterns of SARS-CoV-2 genomes. The amount of clinical and genomic data from patients admitted at ICU in different hospitals is remarkable. However, there are some fundamental information that the authors did not report and that are essential for their scope.

Authors : We thank the reviewer for their positive comments on our work and for the time spent on it.

The first essential information that the author did not mention in their analysis was the vaccination status of the patients involved in the study. During the data collection of the present study (between October 2020 and 30 May 2021), the number of people vaccinated with at list one dose in France increased from 0% to 38,7%. The efficacy of vaccines in preventing mortality is well documented in many works, so the information of vaccination status of patients involved in the study must be considered in the multivariate analysis.

Authors : We agree with the reviewer that reporting the vaccination status of the patients would make a lot of sense. We however did not record this variable because there was no SARS-CoV-2 vaccine available when the inclusion period of the study started and during the majority of the inclusion period (March 2020 – May 2021), as vaccination started in January 2021 in France. Additionally, the efficacy of vaccination on preventing ICU admission was very high on Alpha variant, implying that the vast majority of ICU-admitted patients, even after January 2021, are more likely to be non-vaccinated patients, resulting in a very low percentage of vaccinated patients in the cohort. We have however acknowledged this in the limitations section of the discussion, as follows (lines 348 to 353): « We also did not record SARS-CoV-2 vaccination status as there was no anti-SARS-CoV-2 vaccine available at the time the study started. However, on May 1st 2021, when the inclusion period ended, less than 10% of the French population had been fully vaccinated (https://covidtracker.fr/vaccintracker/), implying that the proportion of vaccinated patients in this cohort of critically ill patients was very low.”

The second essential information that was not reported in this work is the total number of SARS-CoV-2 positive patients. Previous works (like Patone et al., The Lancet Infectious Diseases 2021) observed that patients with variant “alpha” were at increased risk of ICU admission and 28-day mortality; however considering only patients that received critical care, mortality appeared to be independent of virus strain. To better investigate the relationship between variants and clinical outcomes it should be necessary to also evaluate the ratio between the number of deaths and the number of infected people, like Davies et al. Nature 2022, or Funk et al. EUROSURVEILLANCE 2021.

Authors : We understand the reviewer’s point and agree that previous studies, which used large national databases, including those of Patone et al. (Ref #6 of the revised manuscript), Davies et al. (Ref #8), or the added reference of Funk et al. (Ref #7), established a link between the number of infected people and hospital deaths. However, our study design does not allow for such a relationship to be established as we did not rely on a large population-based database. However, we would like to stress that using such a large database would not have allowed for depicting the clinical phenotype of the patients according to variant type nor to perform deep genome sequencing. We have added this limitation to the revised Discussion section of the manuscript, as follows (lines 342 to 344): « We did not use a population-based database, precluding relationships to be studied between the activity of COVID-19 epidemic and risk of ICU death according to variant type. »

Specific comments:

Line 85: which are “the substitutions selected so far”? By whom? Please explain better

Authors : We agree with the reviewer that this sentence should be clarified. We modified it as follows:

“Many of the substitutions that characterize emerging variants (VOC and VOI) are located in the Spike (S) protein, the protein involved in viral attachment and entry into cells and the target of neutralizing antibodies”.

Line 107: Other variants have been also investigated, please cite at list  “Funk et al. EUROSURVEILLANCE 2021”

Authors : we have now added the reference by Funk et al. in the revised introduction section of the manuscript (ref#7 of the revised manuscript).

Lines 159-161 : authors reported the methods of a phylogenetic analysis that is not reported in the results. Please report the SARS-CoV-2 phylogenetic tree: the mutations found with a statistically significant difference are harbored by SARS-CoV-2 genomes that are phylogenetically linked?

Authors : The reviewer is correct. A phylogenetic tree has been added (Supplementary figure 1) showing that the mutation Orf3a-Q57H is mainly harbored by variants from lineages B.1.160 and B.1.351.

Line 186 : please add the list of GISAID IDs of the sequenced samples in supplementary.

Authors : we have now added a supplementary file containing GISAID IDs of the sequenced samples.

Figure 1: please also report the number of samples collected every week. If it not possible to report these data into the same figure, please add another figure as supplementary.

Authors : we agree with the reviewer that this information should be added in the manuscript. A supplementary table has been added accordingly (Supplentary table 1).  

Lines 268-270 : this observation could be inverted: ICU patients infected with variant alpha and were characterized by a younger age and less comorbidities; it is possible to conclude that alpha variant is associated with a severe disease in younger people? Can be better investigated the correlation between mutational patterns and age of ICU admission?

Authors : As suggested by the reviewer, we have now modified the sentence of the first paragraph of the discussion relating to the characteristics of variant alpha. Upon ICU admission, infection with alpha variant was indeed associated with less severe disease, as quantified with SAPS II and SOFA scores, in younger patients. However, as shown in Table 2, there was no impact of alpha variant on day 28 or day 90 mortality.

The aim of this study was not to study the relationship between age and mutational patterns. However, because age is an important prognosis covariable, we have adjusted for age all analyses that assessed the relationship between mutational patterns and mortality (see Table 3).

Round 2

Reviewer 2 Report

Fourati et al. describe a multicenter clinical study with the aim to investigate correlations between clinical severity of COVID-19 and mutational patterns of SARS-CoV-2 genomes. The strength of this study is that they analyzed a remarkable number of clinical and genomic data from patients admitted at ICU in different hospitals. The current revision of this manuscript addresses my concerns with the previous version, and it takes this manuscript to the level it needs to be at for publishing. I would have enjoyed seeing correlations of their observations with vaccination status.